# GENERALIZING POINCARÉ POLICY REPRESENTATIONS IN MULTI-AGENT REINFORCEMENT LEARNING

## ABSTRACT

Learning policy representations is essential for comprehending the intricacies of agent interactions and their decision-making processes. Recent studies have found that the evolution of any state under Markov decision processes (MDPs) can be divided into multiple hierarchies based on time sequences. This conceptualization resembles a tree-growing process, where the policy and environment dynamics determine the possible branches. In this paper, the multiple agent's trajectory growing paths can be projected into a Poincaré ball, which requires the tree to grow from the origin to the boundary of the ball, deriving a new geometric idea of learning **P**oincaré **P**olicy **R**epresentations (P2R) for MARL. Specifically, P2R captures the policy representations of Poincaré ball by a hyperbolic neural network and constructs a contrast objective function that encourages embeddings of the same policy to move closer together while embeddings of different policies to move apart, which enables embed policies with low distortion. Experimental results provide empirical evidence for the effectiveness of the P2R framework in cooperative and competitive games, demonstrating the potential of Poincaré policy representations for optimizing policies in complex multi-agent environments.

## 1 INTRODUCTION

Multi-agent reinforcement learning (MARL) is widely used in a variety of applications, ranging from robotics (Kober et al., 2013) and autonomous vehicles (Shalev-Shwartz et al., 2016) to real-time strategy games (Vinyals et al., 2019), social networks (Chen et al., 2020b), and economic markets (Qiu et al., 2021). In MARL, agents interact with each other and their environment to learn effective policies. One of the key challenges in MARL is learning effective representations of the agents' policies, which need to capture the characters of the policies and dynamics of the system that enable efficient decision-making. Furthermore, policy representations in MARL are crucial to realize cooperation among agents, improve the performance and efficiency of the multi-agent system, and adapt to different tasks and environments.

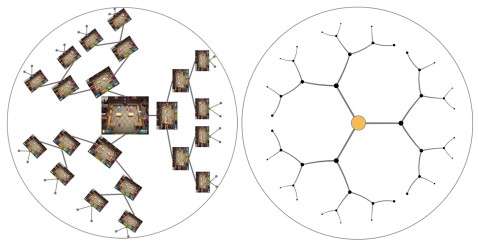

Figure 1: The state relationships of MDPs of *Overcooked* environment could be divided into multiple hierarchies based on time sequences (left), which resembles a tree-growing process projected into a Poincaré ball (right).

Many recent works have been devoted to learning informative representations for agent policies using deep learning architectures for reinforcement learning (RL) (Albrecht & Stone, 2018; Papoudakis et al., 2021; Rabinowitz et al., 2018). He et al. (2016) introduced a novel method that focuses on the learning of a modeling network tasked with reconstructing the actions of a modeled agent based on its observations. Grover et al. (2018) put forward an innovative approach that relies on imitation learning where they train a mapping from observations to actions in a supervised manner to capture a point-based policy representation. Raileanu et al. (2018) contributed to the field by developing an algorithm designed to learn the inference of an agent's intentions by leveraging the policy of the controlled agent. Tacchetti et al. (2018) introduced a notable concept involving relational forward models that utilize graph neural networks for modeling agents. Zintgraf et al. (2021) employed a Variational Autoencoder (VAE) for agent modeling, particularly for fully-observable

tasks. However, the aforementioned policy representation methods assume that the trajectory data structure has Euclidean property with state linear transformation, and we notice that trajectory data has an implicit hierarchical property, which induces tree-like state evolution.

We thus consider a different structure, and further find the evolution of any state of Markov decision processes (MDPs) (Puterman, 2014) can be divided into multiple hierarchies. This conceptualization resembles a tree-growing process, where the policy and environment dynamics determine the possible branches. These hierarchical evolution relationships are nonlinear by the randomness of the environment dynamic and the police, making hierarchy a natural basis to encode information for MARL. In this work, we assume the agent policies are black boxes, that is, we can only access them based on the interaction data with the environment, which we utilize to learn policy representations. Accordingly, learning effective policy representations should prioritize capturing precisely hierarchically-structured features of the trajectories. Specifically, we leverage any state as the root of the tree structure and construct trees, as shown in Figure 1. The growth space and direction of the trees are determined by the action distribution of the policy and the environment randomness at each time step. The trees formed by the trajectories of different policies exhibit distinct characteristics, such as the width and depth of the left and right subtrees. Furthermore, the enclosing geometry of the Poincaré ball is precisely exponential growth from the origin to the boundary, which enables embedding the hierarchical tree-like trajectories with low distortion (Sarkar, 2011)).

In this paper, we model the trajectories of multi-agent interaction with each other and their environment as the growth process of a tree and describe the tree-like trajectory data in a Poincaré ball. We embed the any state of a trajectory in the central region of the Poincaré ball, and as the agent interacts with the environment, the tree grows from the central region toward the edge of the ball. Specifically, we propose a novel framework (P2R) to learn policy representations in Poincaré ball for multi-agents. P2R captures the policy representations of Poincaré ball by a hyperbolic neural network and constructs a contrast objective function that encourages embeddings of the same policy to move closer together while embeddings of different policies to move apart.

**Superiority of Poincaré Policy Representations.** The experimental results demonstrate the effectiveness of the P2R framework in both cooperative and competitive environments. These remarkable findings emphasize the tremendous potential of Poincaré policy representations and illuminate the path to enhancing policy optimization in complex multi-agent environments.

## 2 PRELIMINARIES

**Reinforcement Learning**  The traditional formulation of the reinforcement learning (RL) problem revolves around the concept of a Markov Decision Process (MDP), defined by the tuple $M = (S, A, P, R, \gamma)$ where $S$ and $A$ stand for the state space and action space respectively, while $R(s, a)$ represents the reward function. The transition dynamics, denoted as $P(s'|s, a)$, dictate how the environment's state evolves, and the discount factor $\gamma \in [0, 1)$ quantifies the agent's inclination towards earlier rewards. Within this framework, we introduce a stochastic policy $\pi_\theta$ that depends on a parameter vector $\theta$. The interaction between this policy and the environment leads to the creation of a trajectory $\tau$, which can be expressed as a sequence of state-action pairs: $\tau = \{(s_t, a_t)\}_{t=1}^T$, with $T$ representing the maximum time step in an episode. The agent's ultimate goal is to learn a policy that maximizes its expected discounted accumulative rewards over trajectories:

$$\arg\max_\theta \mathbb{E}_{\tau \sim \pi_\theta, P} \left[ \sum_{t=0}^\infty \gamma^t R(s_t, a_t) \right]. \tag{1}$$

It's important to note that this formulation seamlessly extends to multi-agent scenarios.

**Poincaré Geometry**  A *hyperbolic space* $\mathbb{H}^n$ represents an $n$-dimensional Riemannian manifold with constant negative sectional curvature $-c$. Beltrami (1868b) established the equiconsistency of hyperbolic and Euclidean geometry, introducing the renowned *Poincaré ball model* named after its re-discoverer. The Poincaré ball $(\mathbb{B}^n, g^{\mathbb{B}})$ is defined by the manifold $\mathbb{B}^n = \{x \in \mathbb{R}^n \,|\, \|x\| < 1\}$ equipped with the Riemannian metric tensor $g_x^{\mathbb{B}} = \lambda_x^2 g^E$, where $\lambda_x := \frac{2}{1-\|x\|^2}$ is the *conformal factor* and $g^E$ denotes the Euclidean metric tensor.

The Poincaré ball model provides a *geodesic distance* for vector $x, y \in \mathbb{B}^n$:

$$d_{\mathbb{B}}(x, y) = \cosh^{-1}\left(1 + 2\frac{\|x - y\|^2}{(1 - \|x\|^2)(1 - \|y\|^2)}\right). \tag{2}$$

Hyperbolic geometry does not like the Euclidean vector space which invokes the affine vector operations such as the summation, multiplication, etc. To address this issue, we follow Ganea et al. (2018); Shimizu et al. (2020), and utilize the framework of *gyrovector spaces* as introduced by Ungar (2022) to extend common vector operations into hyperbolic space. The curvature of Poincaré ball is modified by $c$, Poincaré ball is then defined as $\mathbb{B}_c^n = \{x \in \mathbb{R}^n \mid c\|x\|^2 < 1, c \geq 0\}$. The corresponding *conformal factor* takes the form $\lambda_x^c := \frac{2}{1 - c\|x\|^2}$.

**Möbius addition.**  Given a curvature $-c$ of $\mathbb{B}_c^n$ and $x, y \in \mathbb{B}_c^n$, the *Möbius addition* is defined as:

$$x \oplus_c y = \frac{(1 + 2c\langle x, y\rangle + c\|y\|^2)x + (1 + c\|x\|^2)y}{1 + 2c\langle x, y\rangle + c^2\|x\|^2\|y\|^2}. \tag{3}$$

**Exponential and logarithmic maps.**  The *exponential* map $\exp_x^c$ is a function from tangent space $T_x\mathbb{B}_c^n \cong \mathbb{R}^n$ to $\mathbb{B}_c^n$, which provides a way of mapping vectors from the Euclidean space to hyperbolic space. Given $x, y, v \in \mathbb{B}_c^n$, the *exponential* map $\exp_x^c$ is defined by:

$$\exp_x^c(v) := x \oplus_c \left(\tanh\left(\sqrt{c}\frac{\lambda_x^c\|v\|}{2}\right)\frac{v}{\sqrt{c}\|v\|}\right). \tag{4}$$

The reverse process of exponential map is the *logarithmic* map, which is defined as:

$$\log_x^c(y) := \frac{2}{\sqrt{c}\lambda_x^c}\text{arctanh}(\sqrt{c}\| - x \oplus_c y\|)\frac{-x \oplus_c y}{\| - x \oplus_c y\|}. \tag{5}$$

**Parallel transport.**  Let $T_x\mathbb{B}_c^n$ be the tangent space to vector $x \in \mathbb{B}_c^n$, the *parallel transport* establishes a linear isometric mapping between two tangent spaces, that is, relocating a tangent vector $y \in T_{\mathbf{0}}\mathbb{B}_c^n$ to $T_x\mathbb{B}_c^n$ via vector affine operations. This process includes mapping $y$ into $\mathbb{B}_c^n$ through the exponential map, then transitioning to the point $x$ using *Möbius addition* $\oplus_c$, and finally mapping into $T_x\mathbb{B}_c^n$ via logarithmic operation. The *parallel transport* is given by the following isometry:

$$P_{\mathbf{0} \to x}^c(y) = \log_x^c(x \oplus_c \exp_{\mathbf{0}}^c(y)) = \frac{\lambda_{\mathbf{0}}^c}{\lambda_x^c}y. \tag{6}$$

Through *parallel transport*, we can establish a connection between two distinct tangent spaces.

## 3 POINCARÉ POLICY EMBEDDINGS

In this work, we assume that the agent policies are black boxes, that is, our access to the policies is solely through interaction data with the environment. Drawing inspiration from prior research, specifically (Grover et al., 2018) and (Papoudakis et al., 2021), we recognize that trajectories, composed of state-action pairs, inherently convey the policy's characteristics. Leveraging trajectories as a means to acquire policy embeddings proves to be an effective strategy.

To formalize this, our objective is to learn policy embeddings for each agent, denoted as $f_{\mathbf{\Theta}} : E_\alpha \to \mathbb{B}^n$, where $E_\alpha$ represents the space of episode trajectories $\tau_\alpha$ associated with agent $\alpha$ during interactions with other agents and the environment, $n$ signifies the dimensionality of the embeddings, and $\mathbf{\Theta}$ denotes the function's parameters. These trajectories exclusively consist of state-action pairs for agent $\alpha$. Specifically, $\mathbf{\Theta} = \{\Theta_1, ..., \Theta_L\}$ refers to the parameters of the hyperbolic neural network, with each layer $\Theta_l$ encompassing weight parameters $\Theta_l^M$ and bias parameters $\Theta_l^b$. Accordingly, for MARL, we introduce the following auxiliary tasks for learning an agent's policy representation:

- 1. Policy representation. The policy embeddings should possess the capability to capture hierarchical information. We project the trajectories into hyperbolic space and employ hyperbolic fully-connected (FC) neural networks for policy representation learning.

- 2. Policy discrimination. The obtained policy embeddings should be adept at distinguishing an agent's policy from those of other agents. To achieve policy discrimination, we leverage distance metrics within the Poincaré ball. Embeddings acquired from different trajectories but corresponding to the same policy should be proximate in the embedding space, while embeddings for distinct policies should exhibit greater separation.

## 3.1 Obtain policy representations in Poincaré ball

We propose a method for obtaining policy embeddings within the Poincaré ball. To ensure minimal distortion, we first project trajectories into the Poincaré ball, and subsequently, we use a hyperbolic neural network to learn policy representations from the projected trajectories.

We suppose the episode trajectory space $E_\alpha = \{\tau_\alpha^k\}_{k=1}^K$ of agent $\alpha$ comprising $K$ trajectories. Since trajectories are situated within Euclidean space, and we need to learn policy embeddings in a hyperbolic space. To bridge the gap between Euclidean trajectories $\tau_\alpha^k$ with dimensionality $m$ and the hyperbolic space $\mathbb{H}^m$, we employ the exponential map from the origin of the Poincaré ball to project $\tau_\alpha^k$ in hyperbolic space. We denote the result of mapping $\tau_\alpha^k$ into the hyperbolic space as $\hat{\tau}_\alpha^k$, and this procedure is formally defined as follows:

$$\hat{\tau}_\alpha^k = \exp_{\mathbf{0}}^c(\tau_\alpha^k) = \tanh\left(\sqrt{c}\|\tau_\alpha^k\|\right)\frac{\tau_\alpha^k}{\sqrt{c}\|\tau_\alpha^k\|}. \tag{7}$$

Subsequently, we employ hyperbolic neural networks to extract hierarchical relationships and other essential features embedded within these trajectories. This approach ensures a high capacity to capture complex structures and extracts tree-like properties within the hyperbolic space. Specifically, we leverage the Möbius matrix-vector multiplication (Ganea et al., 2018) to define hyperbolic neural networks. Considering the $l$ layer, given the weight parameters $\Theta_l^M \in \mathcal{M}_{n,m}(\mathbb{R}) : \mathbb{R}^m \to \mathbb{R}^n$ is a linear map, which we identify with its matrix representation, then $\forall(\hat{\tau}_\alpha^k) \in \mathbb{B}_c^n$, we have:

$$\Theta_l^M \otimes_c \hat{\tau}_\alpha^k = (1/\sqrt{c})\tanh\left(\frac{\|\Theta_l^M \hat{\tau}_\alpha^k\|}{\|\hat{\tau}_\alpha^k\|}\tanh^{-1}(\sqrt{c}\|\hat{\tau}_\alpha^k\|)\right)\frac{\Theta_l^M \hat{\tau}_\alpha^k}{\|\Theta_l^M \hat{\tau}_\alpha^k\|}. \tag{8}$$

Biases are introduced into the hyperbolic neural networks, and these bias translations of the Poincaré ball are naturally achieved by moving along geodesics. We leverage the *parallel transport* to give the Möbius translation of $\hat{\tau}_\alpha^k \in \mathbb{B}_c^n$ by the $l$ layer bias parameters $\Theta_l^b \in \mathbb{B}_c^n$,

$$\hat{\tau}_\alpha^k \oplus_c \Theta_\alpha^b = \exp_{\hat{\tau}_\alpha^k}^c(P_{\mathbf{0}\to\hat{\tau}_\alpha^k}^c(\log_{\mathbf{0}}^c(\Theta_l^b))) = \exp_{\hat{\tau}_\alpha^k}^c\left(\frac{\lambda_{\mathbf{0}}^c}{\lambda_{\hat{\tau}_\alpha^k}^c}\log_{\mathbf{0}}^c(\Theta_l^b)\right). \tag{9}$$

Finally, the unified form of the hyperbolic neural network encompasses multiple layers and integrates Equation (7), Equation (8), and Equation (9). Mathematically, it can be expressed as:

$$f_{\Theta^l}(\tau_\alpha^k) = \varphi_l(\Theta_l^M \otimes_c \left(\exp_{\mathbf{0}}^c\left(\tau_\alpha^k\right)\right) \oplus_c \Theta_l^b), \tag{10}$$

where $\varphi_l$ represents the pointwise non-linearity of the $l$-th layer. In Equation (10), the trajectory $\tau_\alpha^k$ undergoes a series of operations. Initially, it is mapped into the Poincaré ball using the exponential map. Subsequently, it undergoes a transformation based on weight parameters represented as $\Theta_l^M$. Next, it is further adjusted based on bias parameters denoted as $\Theta_l^b$, and finally, a non-linear transformation is applied through $\varphi_l$. These steps are applied in each layer of the hyperbolic neural network, effectively enabling the network to capture hierarchical features of the trajectories.

## 3.2 Consistent policy representations for each agent

Policy embeddings derived from distinct trajectories of the same policy must maintain consistency, signifying the adherence to common policy characteristics, and should exhibit proximity within the embedding space. To ensure the consistency among policy embeddings obtained from different trajectories of the same policy, we compute the distance between policy embeddings using the distance equation within the Poincaré ball, as defined in Equation (2). Subsequently, we define a consistency

objective function based on this distance measure:

$$\mathcal{L}_{con}\Big(\boldsymbol{\Theta}\Big) = \frac{1}{\mathcal{A}} \sum_{\alpha=1}^{\mathcal{A}} \mathbb{E}_{k \neq k'} \left[ \log\left( 1 + \frac{d_{\mathbb{B}}\Big(f_{\boldsymbol{\Theta}}(\tau_\alpha^k), f_{\boldsymbol{\Theta}}(\tau_\alpha^{k'})\Big)}{\epsilon} \right) + \mu\left( \log\frac{1}{1 - \|f_{\boldsymbol{\Theta}}(\tau_\alpha^k)\|} \right) \right],$$
(11)

where $1 \leq k, k' \leq K$, $\mathcal{A}$ denotes the number of agents, $\epsilon$ serves as an adjustment parameter, and $\mu$ is the regularization coefficient. Employing the logarithm effectively scales down the calculated distances and norms within the policy embedding in the Poincaré ball, resulting in smoother data without altering the fundamental nature of the data or its relationships. The term on the left side of the plus sign is designed to minimize the Poincaré distance between sample points. The denominator includes an adjustment parameter $\epsilon$ to amplify the distance between the two embeddings based on different task data. The term on the right side of the plus sign aims to minimize the embedding point within the bounds of the radius in the Poincaré ball. Placing the norm in the denominator ensures that the result remains positive when taking the logarithm.

By minimizing the distance between embeddings obtained from different trajectories of the same policy, we obtain clusters of distinct embeddings of the same policy within the embedding space. This process accentuates the common characteristics of the policy.

### 3.3 DISCRIMINATIVE REPRESENTATIONS BETWEEN MULTIPLE AGENTS

Policies inherently exhibit diverse action distributions given the same state, leading to distinct characteristics in their trajectories. Policy discrimination stems from the varying action preferences exhibited by different policies, leading to distinctive characteristics in their respective trajectories during interactions with both the environment and other agents. Hence, it becomes imperative for policy embeddings to clearly portray these distinctions among different policies.

These distinctions naturally surface in the embedding space, necessitating that emebddings of dissimilar policies possess well-defined boundaries. To fulfill this criterion, we take measures to achieve that emebddings of distinct policies are significantly separated within the Poincaré ball. Similarly, we employ the distance calculation formula within the Poincaré ball, as defined in Equation (2), to quantify the dissimilarity between policy embeddings generated from trajectories of different agents. Based on this distance measure, we construct the discriminative objective function:

$$\mathcal{L}_{dis}\Big(\boldsymbol{\Theta}\Big) = \frac{1}{\mathcal{A}} \sum_{\substack{\alpha \neq \alpha'}}^{\mathcal{A}} \mathbb{E}_k \left[ \log\left( 1 + \frac{d_{\mathbb{B}}\Big(f_{\boldsymbol{\Theta}}(\tau_\alpha^k), f_{\boldsymbol{\Theta}}(\tau_{\alpha'}^k)\Big)}{\epsilon} \right) \right],$$
(12)

where $\mathcal{A}$ and $\epsilon$ also denote the number of agents and an adjustment parameter, respectively. Within Equation (12), the application of the logarithm operation and the incorporation of $\epsilon$ in the denominator follow the identical rationale as delineated in Equation (11). Therefore, by maximizing the distance between embeddings derived from distinct agents' trajectories, we establish clear boundaries among embeddings of different policies.

### 3.4 ENSEMBLE CONSISTENT-DISCRIMINATIVE REPRESENTATIONS

The consistency objective achieves that policy embeddings derived from different trajectories of the same agent exhibit clustering tendencies in the embedding space. Conversely, the discriminative objective aims to maximize the separation between policy embeddings of different agents within the embedding space. These two objectives complement each other, and we introduce an ensemble approach that integrates both of them. Specifically, to estimate parameters $\boldsymbol{\Theta}$ for policy representation function $f_{\boldsymbol{\Theta}}$, we solve the optimization problem by the total objective function, which combines the above Equation (11) and Equation (12):

$$\mathcal{L}_{ens} = \mathcal{L}_{con} + \beta \mathcal{L}_{dis},$$
(13)

where $\beta$ is a trade-off hyperparameter that controls the relative weights of the consistent and discriminative terms. We train policy representation function by optimizing Equation (13) via stochas-

tic Riemannian optimization methods RSGD (Bonnabel, 2013). The algorithm for the proposed P2R method is presented in Appendix A.1.

## 4 EXPERIMENTS

### 4.1 MULTI-AGENT ENVIRONMENTS

We evaluate the performance of our method P2R in two multi-agent environments (one cooperative, one competitive): *Overcooked* (Carroll et al., 2019), and *Pommerman* (Resnick et al., 2018). These environments impose stringent demands on the sequencing of agent actions, exhibiting a pronounced hierarchical structure in the state evolution. More details about the experiments and additional experiments are presented in Appendix B.

**Cooperative**   The *Overcooked* environment, as shown in Figure 2, is a simplified version of the popular video game *Overcooked* (Ghost Town Games, 2016). Within this environment, two agents assume the roles of chefs in a kitchen tasked with cooking and serving dishes. The kitchen contains only three types of objects: onions (yellow), dishes (white), and a cooking pot (dark grey). The task involves agents placing three onions in the pot, allowing them to cook for a duration of 20 timesteps, transferring the resulting soup into a dish (white), and subsequently serving it (light grey), giving all players a reward of 20. The primary objective is to deliver the soup as many times as possible within the time limit. The agents are equipped with six distinct actions: moving up, moving down, moving left, moving right, taking no action (noop), and "interact," which triggers specific actions based on the tile the player is facing, e.g. placing an onion on a counter. A pivotal aspect of the challenges presented in the *Overcooked* environments is the necessity for agents to possess a keen understanding of their partner's policy characteristics and execute effective coordination accordingly.

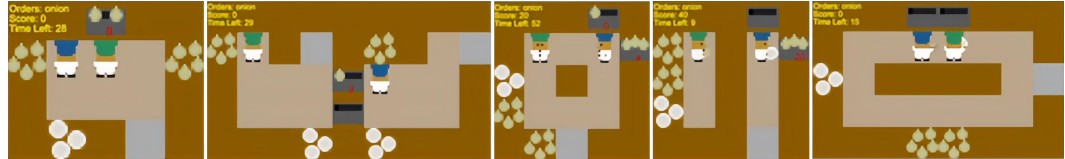

Figure 2: *Overcooked* environment layouts. From left to right: *Cramped Room* confines the agents to a tight space, increasing the likelihood of agent collisions. *Asymmetric Advantages* tests whether agents can devise high-level strategies that capitalize on their individual strengths. In *Coordination Ring*, agents must effectively coordinate their movements to traverse between the bottom left and top right corners of the kitchen. *Forced Coordination* compels agents to formulate a comprehensive joint strategy since neither player can independently serve a dish. *Counter Circuit* involves a non-obvious coordination strategy, where onions are passed over the counter to the pot rather than being carried around. Each layout is equipped with one or more onion dispensers and dish dispensers, providing an unlimited supply of onions and dishes, respectively.

**Pommerman**   The Pommerman environment draws its inspiration from the classic console game Bomberman (W., 1983). In our experiments, we utilize the simulator configured for two agents whose initial positions are randomized close to any of the 4 corners of the board, as depicted in Figure 3. At each time step, each agent has the option to choose from six possible actions: movement in any of the four directions, staying in place, or placing a bomb.

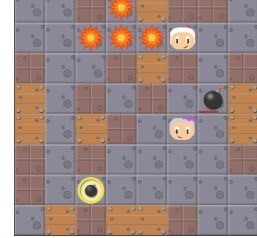

The environment consists of cells that can be passages, rigid walls (dark brown cells), or wood (light brown cells), with maps being randomly generated. Importantly, there is always a guaranteed path between any two agents on the map. The objective of the task is to be the last agent standing, earning a reward of 1 for victory, while tie games result in an episodic reward of -1. When an agent places a bomb, it explodes after 10 time steps, producing flames that last for 2 time steps. These flames have the ability to destroy wood and kill agents within their blast radius. The destruction of wood can reveal either a passage or a power-up (yellow circles). Power-ups fall into three categories: those that increase the blast radius of bombs, those that increase the number of bombs an agent

Figure 3: *Pommerman* environment with $8 \times 8$ board.

can place, and those that grant the ability to kick bombs. An episode of two-player *Pommerman* is finished when an agent dies or when reaching 800 timesteps.

## 4.2 BASELINES

**Local Information Agent Modelling (LIAM) (Papoudakis et al., 2021):** This baseline presents an encoder-decoder agent modeling method capable of extracting concise yet informative representations of modeled agents, relying solely on the local information available to the controlled agent (including its local state observations and past actions). We include LIAM for two reasons: it employs a recurrent encoder for policy representation learning and leverages local state observations in a manner akin to P2R. Through the results, we can compare the performance of memory modeling based on recurrent encoders with tree-like hierarchical modeling in the Poincaré ball.

**Agent Policy Representation Framework (AMF) (Grover et al., 2018):** This baseline introduced an inventive approach centered on imitation learning, where a supervised training scheme is employed to map observations to actions, thereby capturing a point-based policy representation. In contrast to P2R, which learns policy representations in the Poincaré ball, AMF operates in Euclidean space. We can observe whether the policy embeddings within the Poincaré ball, as obtained by P2R, result in better performance. All baselines are trained with PPO algorithm (Schulman et al., 2017).

**Contrastive Agent Representation Learning (CARL):** This baseline is inspired by (Papoudakis et al., 2021) and is a non-reconstruction baseline based on contrastive learning (Oord et al., 2018). CARL utilizes the trajectories of modeled agents during training but restricts execution to solely the trajectories of the controlled agent. Further implementation details for this baseline can be found in Appendix B.4. We included this baseline because the method is a non-reconstructive method that embraces the concept of contrast and employs trajectory-based learning for policy representations.

## 4.3 EXPERIMENT RESULTS

**Cooperative** Figure 4 shows the mean episode rewards of all methods during training in the five *Overcooked* environment layouts. These layouts necessitate specific sequences of actions from the agents and display a state evolution resembling a hierarchical tree-like structure. Our experimental results also validate the effectiveness of our P2R method in learning policy embeddings within the Poincaré ball. It excels at extracting hierarchical information from agent trajectories, resulting in more effective policy embeddings. Furthermore, these policy embeddings encapsulate a richer set of information, which proves advantageous for the decision-making processes of agents.

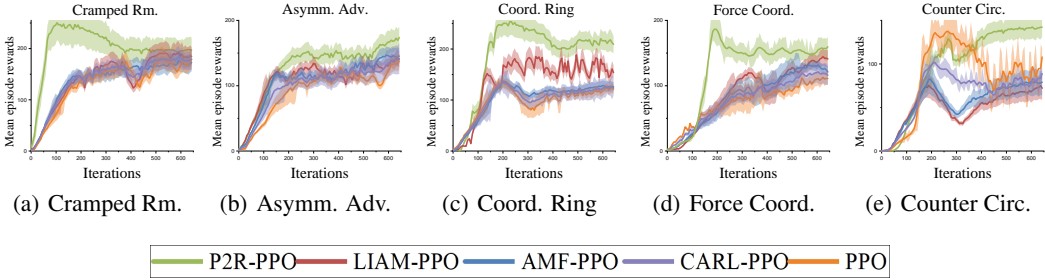

(a) Cramped Rm.  (b) Asymm. Adv.  (c) Coord. Ring  (d) Force Coord.  (e) Counter Circ.

P2R-PPO —— LIAM-PPO —— AMF-PPO —— CARL-PPO —— PPO

Figure 4: Average episode rewards on each layout of the *Overcooked* environment during training, shaded regions indicate the standard deviation over five training seeds.

*Cramped Room*: In this layout, two agents are prone to collisions in the confined space. If agents perceive their teammate's behavioral characteristics, they will adjust their strategies based on their teammate's actions to minimize collisions while completing the task. As shown in Figure 4(a), P2R-PPO typically results in one agent learning to complete the task faster in the initial approximately 100 iterations of the experiment while the other agent remains inactive, causing minimal interference. The P2R policy embeddings quickly learn the characteristics of the teammate's policy, and both agents operate without interference from each other, resulting in high scores in the initial stages. As the training process progresses, the previously inactive agent also attempts to complete the task, leading to collisions in the confined space, reducing task efficiency. The P2R policy embeddings

provide behavioral features, such as action execution sequences, for both agents, encouraging faster cooperation. In contrast, other baselines exhibit a more stable learning process, but even in the initial stages, significant interference between the two agents occurs.

*Asymmetric Advantages*: In this layout, two agents need to learn how to allocate the use of the two pots. If both agents only choose one pot, it leads to inefficient task completion. Therefore, the policy embeddings need to include sufficient hierarchical action information, including the order in which agents select pots, the frequency of pot usage, and time allocation, among others. As shown in Figure 4(b), P2R-PPO learn policy embeddings in the Poincaré ball demonstrate better performance.

*Coordination Ring*: The training process in this layout is similar to that in the *Cramped Room*, as depicted in Figure 4(c). Our P2R-PPO method learns policy embeddings that extract more information about action hierarchy and action sequence relationships, enabling the agents to quickly learn to avoid collisions and efficiently complete tasks.

*Forced Coordination*: In this layout, the right agent handles two pots and soup delivery while the left agent is responsible for providing onions and plates, the essence of the challenge lies in the coordination of action execution sequences between the two agents. Moreover, due to the limited positions at the central interaction counter (only three positions), the left agent needs to provide the corresponding raw materials based on the right agent's soup delivery, and the right agent must adjust the order of cooking and delivery based on the sequence of raw materials provided by the left agent. As shown in Figure 4(d), P2R-PPO demonstrates better performance that enables the agents to learn to cooperate more quickly. Furthermore, the agents even learn a form of "laziness" to some extent during training. For a period of time, the left agent places multiple onions and plates on the counter, and the right agent takes them one by one, causing both agents to constantly shuttle between their respective corridors. However, the left agent eventually learns to place one onion or plate at a time in the grid next to the pot, and the right agent learns to take raw materials only from the nearest grid. This is because P2R's policy embeddings effectively capture the hierarchical characteristics of the behavior of the agent on the right side, and the agent on the left side has learned to "slack off" by utilizing the information provided by the policy embeddings.

*Counter Circuit*: In this layout, the most efficient cooperative policy involves passing onions through the central counter instead of both agents continuously circling the ring. As illustrated in Figure 4(e), the two agents using our P2R policy embeddings learn each other's action hierarchy characteristics, significantly improving cooperation efficiency.

**Competitive**    In *Pommerman* experiment, agents move to positions that could potentially eliminate their opponents by placing bombs and then quickly retreating. This results in repeated action sequences that exhibit clear hierarchical action patterns, and the evolution of states also demonstrates a tree-like hierarchical structure. To test the effectiveness of policy embeddings, we combine the opponent's embeddings with the current state as the input for the agent. If the learned opponent's policy embeddings are effective, the agent, when making action selections, will take into account the characteristics of the opponent's policy. If the opponent's policy is ineffective, the current agent cannot obtain information about the opponent, which will directly reflect in the win rate results. We conducted two sets of experiments: (1) comparing the win rates of all baselines combined with PPO against the naive PPO algorithm and (2) comparing the win rates between all baselines

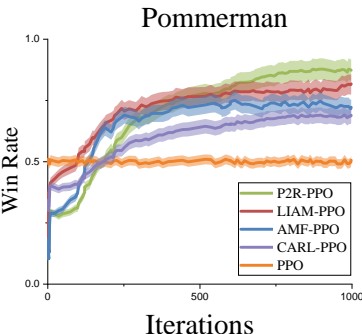

Figure 5: Average win rates of baseline agents against naive PPO agents across five training seeds in the *Pommerman* environment.

(include naive PPO) after training for 1000 iterations, and all baselines are combined with PPO. In terms of winning rates, the combination of the PPO algorithm with policy embeddings learned by P2R achieved the highest winning rates in both sets of experiments. This demonstrates that learning policy embeddings within the Poincaré ball is more effective at capturing hierarchical information.

The average win rates of baseline agents against naive PPO agents across five training seeds are shown in Figure 5. In the initial stages of the training process, policies are busy exploring the environment and have not yet exhibited hierarchical state evolution characteristics.

Therefore, there is not enough information provided for P2R to learn hierarchical characteristics. As the training process progresses, all baseline methods effectively learn the characteristics of the opponent's policy, with P2R method showing superior performance. Moreover, agents in adversarial matches tend to exhibit certain patterns, meaning that policies reveal some habitual action sequences. When P2R method capture such characteristics, it outperforms other methods when combined with PPO, resulting in higher win rates. The win rates between all baselines (include naive PPO) after training for 1000 iterations are shown in Figure 6. By comparing the win rates between each baseline and P2R, we observe that our P2R method achieves higher win rates (over 50%) against other algorithms. In competitive environments where action

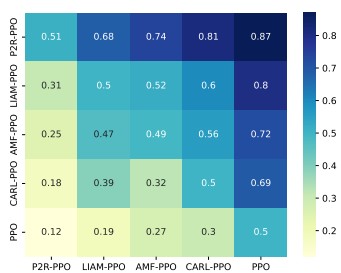

Figure 6: Win rates of baseline agents in adversarial matches after 1000 iterations in the *Pommerman* environment.

sequences exhibit strong hierarchical characteristics and state evolution resembles tree-like growth, learning policy embeddings in the Poincaré space proves effective in capturing the hierarchical features of policies, thereby enhancing the decision-making process.

## 4.4 EMBEDDING ANALYSIS

We qualitatively visualize the policy embeddings learned by P2R via HoroPCA (Chami et al., 2021). As shown in Figure 7 for 10 test interaction episodes of 5 randomly selected agents in the initial cases and after trained cases of the *Forced Coordination* of *Overcooked* and *Pommerman*, respectively. The policy embedding visualizations of other environments are in Appendix B.2.

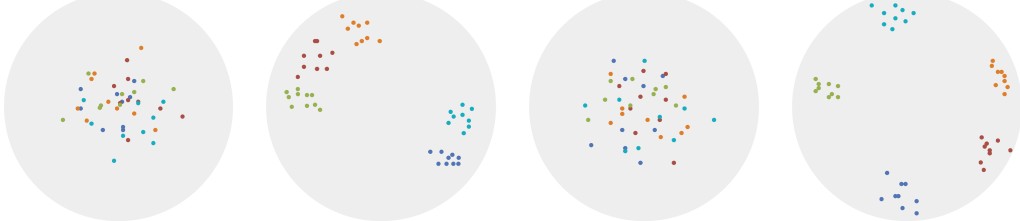

(a) Forced Coord. initial    (b) Forced Coord. trained    (c) Pommerman initial    (d) Pommerman trained

Figure 7: Policy embeddings obtained by P2R for 10 test episodes involving 5 randomly selected agents are visualized using HoroPCA for two different environments. Each color represents a distinct agent policy. Intuitively, policy embeddings of the same agent tend to cluster together in space, while those of different agents are dispersed, indicating that P2R effectively captures diverse policy features and exhibits strong discriminative power in policy representation.

The initial policy embeddings in the Poincaré ball are scattered within a certain region. After training, the P2R method distinctly separates the ten policy for each of the five agents into five clusters.

## 5 CONCLUSION

In conclusion, this paper presents a novel framework for multi-agent reinforcement learning (MARL) based on a new geometric policy representation perspective from the non-Euclidean hyperbolic projection. By leveraging the hierarchical structure inherent in Markov decision processes (MDPs), our approach projects the trajectories of multiple agents onto a Poincaré ball, enabling policy representations that are both efficient and effective. Our key innovation lies in modeling the policy representation as a tree-growing process from the centric Poincaré ball to its boundary. To enhance this hierarchical property for further geometric generalization, we design a contrastive objective function that encourages consistent policies to be embedded closer together in the hyperbolic space, while pushing inconsistent policies farther apart. This leads to representing those policies with low distortion using only a few dimensions, demonstrating the geometric expression of hyperbolic embeddings in MARL. Experimental results showcase the superiority of our P2R framework over state-of-the-art methods across cooperative and competitive games. These findings emphasize the potential of non-Euclidean policy representations for improving the performance and scalability of control policies in complex multi-agent environments.

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

# A APPENDIX

## A.1 ALGORITHM

---

**Algorithm 1** P2R

---

**Require**: number of policies $\mathcal{A}$, batch size $H$ for each policy, max iteration $Q$, period to train policy representation module $U$.

**Initialize**: $\mathcal{A}$ policies $\{\pi_\alpha\}_{\alpha=1}^{\mathcal{A}}$, parameters of policy representation module: $\Theta$, environment $E_{nv}$, policy experience replay buffer $\{D_1, \cdots, D_{\mathcal{A}}\}$, representation experience replay buffer $\{B_1, \cdots, B_{\mathcal{A}}\}$, representation buffer $F$, policy embeddings $\{\hat{\pi_1}, \cdots, \hat{\pi_\alpha}, \cdots, \hat{\pi_{\mathcal{A}}}\}$.

---

1: **while** $q < Q$ **do**
2:     Reset the environment $E_{nv}$
3:     **for** $\alpha = 1, 2, \cdots, \mathcal{A}$ policies **do**
4:         Get teammate's (or opponent's) policy embedding $\hat{\pi_{\alpha'}}$ for $\pi_\alpha$ via Equation (10).
5:         **for** each episode running time step t **do**
6:             Get state $s_t^\alpha$ from environment $E_{nv}$ for policy $\pi_\alpha$.
7:             Sample action $a_t^\alpha$ from policy $\pi_\alpha(a_t^\alpha|s_t^\alpha, \hat{\pi_{\alpha'}})$.
8:             Apply the action $a_t^\alpha$ to the environment.
9:             Get next state $s_{t+1}^\alpha$ and the reward $r_t^\alpha$.
10:            Store the transition $(s_t^\alpha, a_t^\alpha, r_t^\alpha, s_{t+1}^\alpha)$ into policy experience replay buffer $D_\alpha$.
11:            Store the transition $(s_t^\alpha, a_t^\alpha, r_t^\alpha, s_{t+1}^\alpha)$ into representation experience replay buffer $B_\alpha$.
12:            **if** environment $E$ done **then**
13:                Continue
14:            **end if**
15:         **end for**
16:         **if** size of replay buffer $D_\alpha$ greater than batch size $H$ **then**
17:            Update policies $\pi_\alpha$ with policy objective function.
18:            Clear policy experience replay buffer $B_\alpha$.
19:            Continue
20:         **end if**
21:     **end for**
22:     **if** $q \mod U == 0$ **then**
23:         **for** $\alpha = 1, 2, \cdots, \mathcal{A}$ policies **do**
24:            **for** $k = 1, 2, ..., K$ trajectories in replay buffer $D_\alpha$ **do**
25:                Compute the policy embeddings $\{\hat{\pi_\alpha^k}\}_{\alpha=1, k=1}^{\mathcal{A}, K}$ via Equation (10).
26:                Store the policy embeddings $\{\hat{\pi_\alpha^k}\}_{\alpha=1, k=1}^{\mathcal{A}, K}$ into representation buffer $F$.
27:            **end for**
28:            Clear policy experience replay buffer $B_\alpha$.
29:         **end for**
30:         Compute $\mathcal{L}_{ens}$ via Equation (13) with the policy embeddings in representation buffer $F$.
31:         Update $\Theta$ for policy representation module.
32:         Clear representation buffer $F$.
33:     **end if**
34: **end while**

---

The pseudocode for our P2R method is presented in Algorithm 1. In the initial phase, we start with several policies $\mathcal{A}$ and define parameters for the policy representation module. We also set up environment memory buffers to store experiences for both policies and representations and initial policy embeddings. During the training loop, We iterate through training steps until we reach a specified maximum $Q$. For each policy, we interact with the environment to gather experiences. At each step, the policy selects an action based on its current state and its own policy representation. These experiences are stored in a replay buffer. If the buffer size exceeds a certain threshold (batch size), we update the policy to improve its performance. For every $U$ iteration, we focus on improving the policy representations. Specifically, we compute and store the policy embeddings for each policy using the experiences. Then, we calculate a total loss based on these embeddings. This loss guides us

in updating the policy representation module parameters to create better embeddings. We continue this loop until we have completed the maximum number of iterations $Q$.

The policy representation module updates the interval $U$, which is variable and adaptive. Initially, the interval $U$ is kept short because policy changes are frequent at the outset of training, necessitating frequent updates. As the algorithm progresses and policies become more stable, the update interval can be extended. Since policy characteristics evolve during the iterative training process, it is essential to train using the most recent data. To achieve this, we clear the representation experience replay buffer upon completing each policy representation training iteration. This ensures that the training of policy representations always utilizes the most up-to-date data collected from the evolving policies.

## A.2 REINFORCEMENT LEARNING TRAINING

The Poincaré policy embedding $\hat{\pi}_\alpha^k$ of agent $\alpha$ can be used to condition the RL optimized policy. Consider the augmented space $\mathcal{S}_{aug} = \mathcal{S} \times \hat{\Pi}$, where $\mathcal{S}$ is the original observation space of the controlled agent in the P2R-PPO, and $\hat{\Pi}$ is the representation space about the agent's policies. The advantage of learning the policy on $\mathcal{S}_{aug}$ compared to $\mathcal{S}$ is that the policy can specialize for different $\hat{\pi} \in \hat{\Pi}$. The input to the actor and critic is the local observation and the generated policy embedding. We do not back-propagate the gradient from the actor-critic loss to the parameters of the policy representation module. We use different learning rates for optimizing the parameters of the networks of RL and the policy representation module. We empirically observed that P2R exhibits high stability during learning, allowing us to use a larger learning rate compared to RL. Based on the general reinforcement learning objective Equation (1), we use the policy embeddings as the policy condition to optimal policy for agent $\alpha$:

$$\arg\max_{\theta} \mathbb{E}_{\tau \sim \pi_\theta, P} \left[ \sum_{t=0}^{\infty} \gamma^t R((s_t, \hat{\pi_{\alpha'}}), a_t) \right], \tag{14}$$

where $\hat{\pi_{\alpha'}}$ denotes the policy embedding of agent $\alpha'$, which could be the teammate's policy embeddings in the cooperative environments and also could be the opponent's policy embeddings (if available) in the competitive environments.

In our experiments, we optimized the policy of the controlled agent using Proximal Policy Optimization (PPO) (Schulman et al., 2017), however, other reinforcement learning algorithms could be used in its place.

## A.3 RELATED WORK

**Policy representation of MARL**    Learning policy representations is crucial to understanding the emergence of complex phenomena of agents in multi-agent reinforcement learning. A generative method is proposed in (Grover et al., 2018), which proposed an innovative approach that relies on imitation learning where they train a mapping from observations to actions in a supervised manner to capture a point-based policy representation. Two meta-learning methods are proposed in (Rusu et al., 2018; Ghosh & Bellemare, 2020), and they both regard the latent generative representation of learning model parameters as the policy representation, and the method in (Ghosh & Bellemare, 2020) shows more stable performance. (Tacchetti et al., 2018) proposed relational forward models to model agents using graph neural networks. (Zintgraf et al., 2021) uses a VAE for agent modeling for fully-observable tasks. (Rabinowitz et al., 2018) proposed the Theory of mind Network (TomNet), which learns embedding-based representations of modeled agents for meta-learning. (Cetin et al., 2022) show that performance improvements of RL algorithms correlate with the increasing hyperbolicity of the discrete space spanned by their latent representations and also validates the importance of appropriately encoding hierarchical information.

**Hyperbolic representation**    Hyperbolic geometry, as introduced by Beltrami (Beltrami, 1868a) and further developed by Cannon (Cannon et al., 1997), offers a compelling framework for modeling hierarchically-structured features and capturing non-linear relationships, exemplified by the Poincaré ball. Hyperbolic space has found application in various embedding tasks (Kleinberg, 2007; Walter, 2004; Shavitt & Tankel, 2008; Krioukov et al., 2009; Cvetkovski & Crovella, 2009; Krioukov

et al., 2010; Bläsius et al., 2018). Building upon this foundation, Nickel et al. (Nickel & Kiela, 2017) proposed learning Poincaré embeddings for symbolic data, emphasizing latent hierarchical structures. Dhingra et al. (Dhingra et al., 2018) extended these ideas to text and sentence embedding using the Poincaré model, eliminating the need for a projection step through re-parametrization. Tifrea et al. (Tifrea et al., 2018) innovatively embedded words within a Cartesian product of hyperbolic spaces, drawing theoretical connections to Gaussian word embeddings and Fisher geometry. Addressing numerical instability concerns, Yu et al. (Yu & De Sa, 2019) developed a method capable of efficiently representing any point with a small, fixed bounded error within hyperbolic networks.

# B ADDITIONAL EXPERIMENTS

## B.1 ABLATION STUDY

### B.1.1 ABLATE MODULE

To evaluate the contributions of the key modules of the P2R algorithm to its overall performance, we conducted the following ablation study. We systematically removed the consistent policy representation module and the discriminative representation module separately from the method. We then compared the performance differences between the modified methods and the complete method to assess the contributions of these two modules to the method's performance.

We conducted experiments in both the Overcooked environment with its five layouts and the Pommerman environment. In Overcooked, we compared the cooperation scores of the algorithms with each module removed after 650 algorithm iterations and updates. In Pommerman, we compared the adversarial win rates of the algorithms with each module removed after 1000 algorithm iterations and updates.

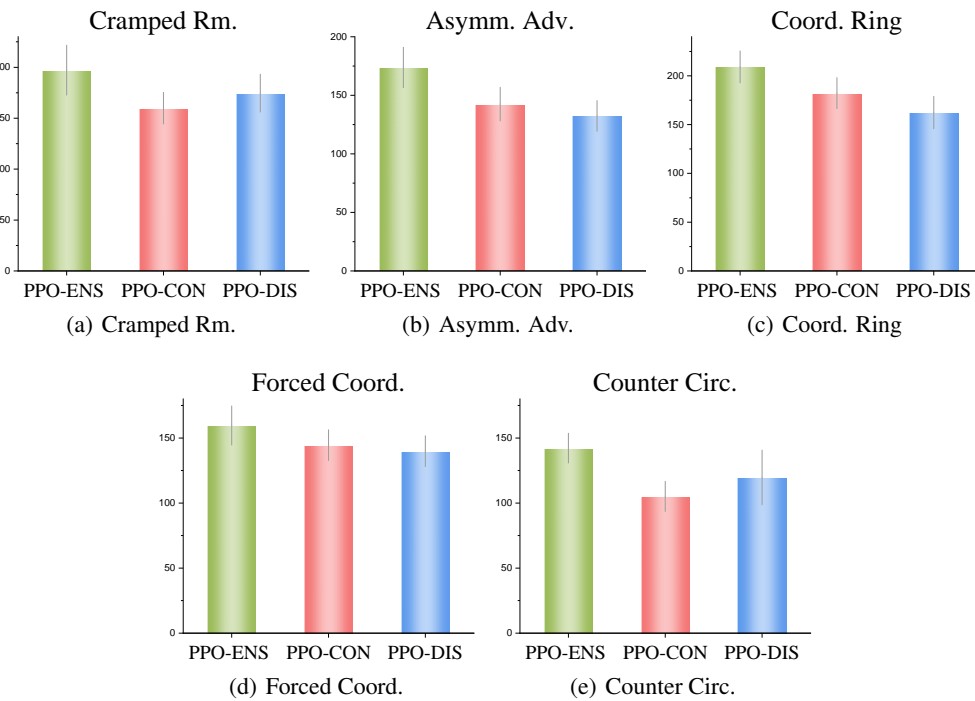

Figure 8: Ablation Study results of the five *Overcooked* environment layouts.

The experimental results are shown in Figure 8. In our ablation study, we still used the PPO algorithm as the policy training algorithm, meaning that all the modified algorithms after removing specific modules were trained in combination with the PPO algorithm. PPO-CON represents the method where the discriminative representation module is removed, but the consistent policy rep-

resentation module is retained. PPO-DIS, on the other hand, represents the method where the consistent representation module is removed, but the discriminative policy representation module is retained. PPO-ENS represents the complete P2R method combined with PPO.

In all of the experimental environments, the PPO-ENS method achieved the best results, indicating that the consistent policy representation module and the discriminative policy representation module are indeed complementary, and their combined use yields the best performance.

However, there is a slight variation in performance. In the *Asymmetric Advantages* layout (Figure 8(b)), *Coordination Ring* layout (Figure 8(c)), and *Forced Coordination* layout Figure (8(d)), the PPO-CON method, which eliminates the discriminative policy representation module, performed relatively well. On the other hand, in the *Cramped Room* layout (Figure 8(a)) and *Counter Circuit* layout (Figure 8(e)), the PPO-DIS method, which eliminates the consistent policy representation module, performed better. Upon analyzing these differences, we found that the effectiveness of the modules depends on the flexibility of policies. In environments where policies have more room for diverse strategies, retaining the discriminative policy representation module (PPO-DIS) is more effective, highlighting its greater role in such scenarios. Conversely, in environments where policy flexibility is limited and diverse strategies are difficult to generate, retaining the consistent policy representation module (PPO-CON) is more effective, indicating its greater impact in such environments. Additionally, these experimental results align closely with the results of policy embedding visualization B.2.

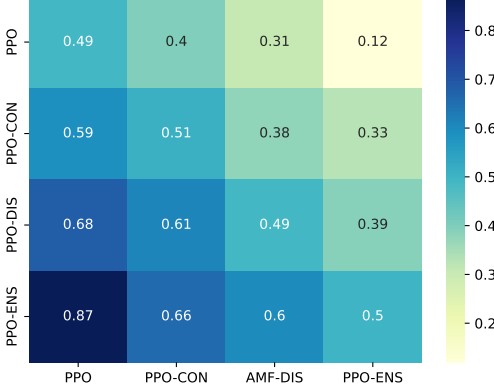

Figure 9: PPO-ENS, PPO-DIS, PPO-CON, and PPO win rates in adversarial matches after 1000 iterations in the *Pommerman* environment.

In the *Pommerman* environment, the mutual win rates among the algorithms are depicted in Figure , the PPO-ENS algorithm achieved higher win rates compared to both PPO-CON and PPO-DIS. This finding similarly demonstrates that in competitive environments, the consistent policy representation module and discriminative policy representation module complement each other. Additionally, in the *Pommerman* environment, we observed that the performance of PPO-DIS was superior to PPO-CON. This is because the *Pommerman* environment falls into the category of environments with a large policy space, meaning it can easily generate a variety of distinct styles of policies. Thus, the discriminative policy representation module plays a more significant role, aligning with the results observed in the *Cramped Room* layout and *Counter Circuit* layout of the *Overcooked* environment.

### B.1.2  EMBEDDING ANALYSIS

To evaluate the robustness of the policy embeddings learned by P2R, we compute multiple embeddings for each policy based on different episodes of interaction at test time. Inspired by (Grover et al., 2018), our evaluation metric is based on the intra- and inter-cluster *geodesic distance* in Poincaré ball between embeddings. The *geodesic distance* in Poincaré ball is defined in Equation (2). The intra-cluster *geodesic distance* for an agent is the average pairwise distance between its embeddings computed on the set of test interaction episodes involving the agent. Similarly, the inter-cluster

*geodesic distance* is the average pairwise distance between the embeddings of an agent with those of other agents. Let $T_\alpha = \{\tau_\alpha^k\}_{k=1}^{K_i}$ denote the set of test interactions involving agent $\alpha$. We define the intra-inter cluster ratio in Poincaré ball (CRPB) based on *geodesic distance* as:

$$
\text{CRPB} = \frac{\frac{1}{\mathcal{A}} \sum_{\alpha=1}^{\mathcal{A}} \frac{1}{K_i^2} \sum_{k \neq k'}^{K_i} d_{\mathbb{B}}(\tau_\alpha^k, \tau_\alpha^{k'})}{\frac{1}{\mathcal{A}(\mathcal{A}-1)} \sum_{\alpha \neq \alpha'}^{n} \frac{1}{K_i} \sum_{k=1}^{K_i} d_{\mathbb{B}}(\tau_\alpha^k, \tau_{\alpha'}^k)},
\tag{15}
$$

where $\mathcal{A}$ is the number of policies. CRPB measures the ratio between the average pairwise distances within clusters of trajectories and the average pairwise distances between different clusters of trajectories.

The numerator calculates the average pairwise distances within clusters. For each policy (indexed by $\alpha$), it computes the average pairwise *geodesic distances* between the trajectories within that policy's set of test episode trajectories. These distances are calculated between all pairs of trajectories within the same policy. The numerator then takes the average of these average distances across all policies.

The denominator calculates the average pairwise distances between different clusters. It considers all possible pairs of policies (indexed by $\alpha$ and $\alpha's$) where $\alpha'$ is the other agent except agent $\alpha$. For each pair of policies, it computes the average pairwise *geodesic distances* between the trajectories of policy $\alpha$ (indexed by $k$) and the trajectories of policy $\alpha'$ (also indexed by $k$). The denominator then takes the average of these average distances across all possible pairs of policies.

In simpler terms, CRPB quantifies how closely related the trajectories within each policy (intra-cluster) are compared to the distances between different policies (inter-cluster) in the Poincaré ball. A higher CRPB value indicates that trajectories within the same policy are more similar to each other than they are to trajectories from other policies, suggesting that the policy embeddings in the Poincaré ball effectively cluster similar policies together. A ratio less than 1 suggests that there is a signal that identifies the agent, and the signal is stronger for lower ratios.

Table 1: Intra-inter clustering ratios in Poincaré ball (CRPB) for *Overcooked* environment five layouts and *Pommerman* environment.

|  | P2R-CON | P2R-DIS | P2R-ENS |
|---|---|---|---|
| Cramped Rm. | 0.72 | 0.58 | 0.22 |
| Asymm. Adv. | 0.82 | 0.63 | 0.43 |
| Coord. Ring | 0.85 | 0.70 | 0.50 |
| Forced Coord. | 0.79 | 0.66 | 0.32 |
| Counter Circ. | 0.76 | 0.54 | 0.29 |
| Pommerman | 0.75 | 0.52 | 0.25 |

The intra-inter clustering ratios are reported in Table 1. The experimental results demonstrate that PPO-ENS has the lowest CRPB values in all environments, indicating its strong ability to identify the agent, which is consistent with the results and conclusions in Section B.1.1. Moreover, when comparing the CRPB values of PPO-ENS in various environments, it is lowest in the *Cramped Room* layout and highest in the *Coordination Ring* layout. This result is also consistent with the findings from Section 4.4, where the policy embeddings in the *Cramped Room* layout exhibit excellent clustering with clear boundaries, making it easier to identify the agent. Conversely, in the *Coordination Ring* layout, the boundaries of policy embeddings are somewhat blurred, and even some outliers appear.

P2R-DIS has lower CRPB values in all environments compared to P2R-CON. This is because the calculation Equation (15) determines that the discriminative representation module's effect is slightly stronger than the consistent representation module. During training, different policy embeddings of the same agent reflect the characteristics of that policy, leading to a natural clustering effect in space. Whether there is a discriminative representation module determines the dispersion

of policy embeddings in space. Additionally, in this formula, as long as the denominator is large enough, it can to some extent limit the value of CRPB.

## B.2 EMBEDDING VISUALIZATION

In Section 4.4, we chose two environments, *Forced Coordination* of the *Overcooked* from the co-operative category and *Pommerman* from the competitive category, to showcase policy embedding visualizations. In this section, we provide a comprehensive display of the visualization results for the five layouts in the *Overcooked* and the *Pommerman* environment (which includes the results presented in Section 4.4).

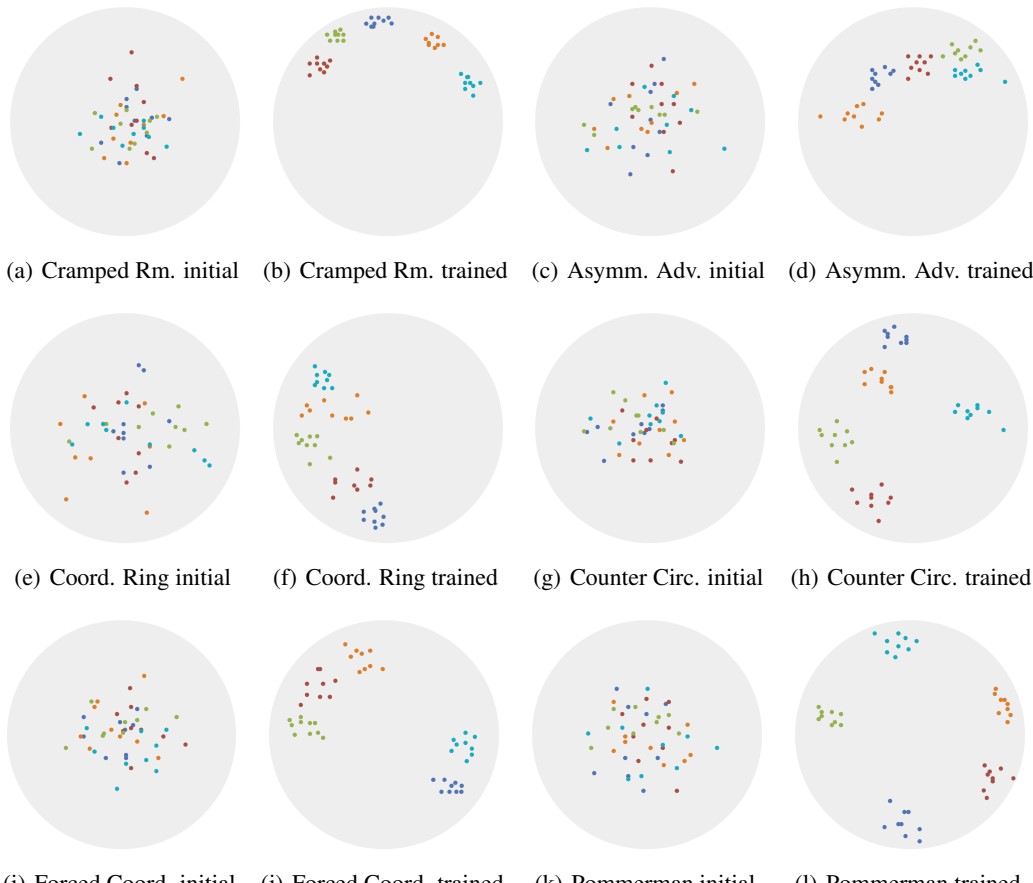

(a) Cramped Rm. initial  (b) Cramped Rm. trained  (c) Asymm. Adv. initial  (d) Asymm. Adv. trained

(e) Coord. Ring initial  (f) Coord. Ring trained  (g) Counter Circ. initial  (h) Counter Circ. trained

(i) Forced Coord. initial  (j) Forced Coord. trained  (k) Pommerman initial  (l) Pommerman trained

Figure 10: Policy embeddings obtained by P2R for 10 test episodes involving 5 randomly selected agents are visualized using HoroPCA for five *Overcooked* layouts and *Pommerman* environment. Each color represents a distinct agent policy. Intuitively, policy embeddings of the same agent tend to cluster together in space, while those of different agents are dispersed, indicating that P2R effectively captures diverse policy features and exhibits strong discriminative power in policy representation.

The visualizations of the six initial policy embeddings and the visualizations of the policy embeddings after training in the six environments are shown in Figure 10. We can observe that, overall, policy embeddings learned under the P2R algorithm tend to exhibit good dispersion in the Poincaré space. They effectively cluster the policy embeddings of the current policy within a specific region in the space. Specifically, in the *Cramped Room* layout (Figure 10(b)), *Counter Circuit* layout (Figure 10(h)), and *Pommerman* environment (Figure 10(l)), the clusters of policy embeddings are notably dispersed, and their boundaries are more distinct. These results indicate that in environments where diverse policy features are easily generated, particularly in those with a highly pronounced hierarchical structure and deeper, broader state evolution hierarchies, the policy embeddings learned by the P2R algorithm tend to form stronger clusters in the space. Conversely, if the environment lacks

such strong hierarchical characteristics, the performance of P2R may weaken, resulting in slightly weaker clustering in the space and occasional outliers among the policy embeddings.

### B.3 EXPERIMENT DETAILS

For all the baselines, we employ Proximal Policy Optimization (PPO) (Schulman et al., 2017) to train 10 agents with 5 different seeds in each environment, that is, 2 agents use one seed. P2R, AMF, and CARL, which are utilized for learning policy representations, are aligned with the iterative nature of policy optimization. Given that each iteration of policy updating may introduce alterations in its characteristics, we integrate policy representation learning alongside these iterations. To be specific, we acquire policy embeddings at periodic intervals during the course of policy updating under the assumption that within these intervals, the policy's characteristics remain relatively stable. Subsequently, these learned policy embeddings can be employed to condition the reinforcement learning-optimized policy.

To elaborate, at each time step when making decisions, the policy combines the current state with the corresponding policy representation, creating an input for the policy network. The LIAM method generates latent variables denoted as $z$ at each time step, augmented with the observation data of the controlled agent, which can also serve as conditioning factors for the reinforcement learning optimized policy.

### B.4 CONTRASTIVE AGENT REPRESENTATION LEARNING

In this section, we introduce the Contrastive Agent Representation Learning (CARL) algorithm, which is a baseline algorithm in this work, mentioned in Section 4.2, and the implementation is similar to (Papoudakis et al., 2021).

CARL employs an approach to extract policy embeddings from agent trajectories within the environment, and it does so without relying on reconstruction. Instead, it leverages local information available to controlled agents, such as their current state and previous actions. During training, CARL has access to the trajectories of all agents in the environment. Still, during execution, it only relies on locally available trajectories.

To extract these policy embeddings, CARL employs a self-supervised learning framework, drawing from recent advancements in contrastive learning (Oord et al., 2018; He et al., 2020; Chen et al., 2020a). Let's consider a scenario with $\mathcal{A}$ agents and a batch of $K$ global episodic trajectories denoted as $\{\tau_{glo}^k\}_{k=1}^K$. Each global trajectory consists of the trajectory of the controlled agent $\alpha$ and the trajectories of all other agents $\alpha'$, given by $\tau_{glo}^k = \{\tau_\alpha^k, \tau_{\alpha'}^k\}$.

Positive pairs are defined within each episode $k$ in the batch between the controlled agent's trajectory and another agent's trajectory:

$$
\begin{aligned}
\text{pos} &= \{\tau_\alpha^k, \tau_{\alpha'}^k\} \text{ (for all } \alpha \neq \alpha' \text{ in the batch)}, \\
\text{neg} &= \{\tau_\alpha^k, \tau_\alpha^{k'}\} \text{ (for all } k \neq k' \text{ in the batch)}.
\end{aligned}
\tag{16}
$$

Two encoders are assumed to exist within CARL: a recurrent encoder that sequentially processes the controlled agent's trajectories $f_{\boldsymbol{w}} : \tau_\alpha \to \Pi_\alpha$, generating the policy embedding $\hat{\pi}_\alpha$, and another recurrent encoder that processes the trajectories of other agents $f_{\boldsymbol{u}} : \tau_{\alpha'} \to \Pi_{\alpha'}$, producing the policy embeddings $\hat{\pi}_\alpha'$, $\boldsymbol{w}$ and $\boldsymbol{u}$ are the parameters of the two recurrent encoder, respectively. The policy embedding $\hat{\pi}_\alpha$ is utilized as input for both the actor and critic components of the Proximal Policy Optimization (PPO) algorithm.

During training, given a batch of episode trajectories, CARL constructs positive and negative pairs as defined in Equation (16). It then minimizes the InfoNCE loss (Oord et al., 2018), which encourages positive pairs to be close and negative pairs to be distant in the embedding space:

$$
\mathcal{L}_{CARL} = -\sum_{\alpha \neq \alpha'}^{\mathcal{A}} \log \frac{\exp\{\cos(\hat{\pi}_\alpha^k, \hat{\pi}_{\alpha'}^k)/\kappa\}}{\sum_{k \neq k'}^K \exp\{\cos(\hat{\pi}_\alpha^k, \hat{\pi}_\alpha^{k'})/\kappa\}},
\tag{17}
$$

where cos represents the cosine similarity, and $\kappa$ controls the temperature of the softmax function, influencing the contrastive loss.

## C  HYPERBOLIC GEOMETRY

### C.0.1  GEODESICS

In the context of the Poincaré ball model, *geodesics* are the shortest paths or curves between two points within hyperbolic space, as shown in Figure C.0.1. The Poincaré ball is a representation of hyperbolic geometry, which is characterized by its intrinsic curvature and differs from Euclidean geometry, where space is flat.

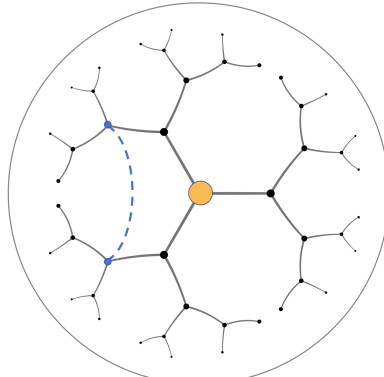

Figure 11: Geodesics (the blue line) on the Poincaré ball model and the shortest curve between two points, analogous to a straight line in the Euclidean space.

In hyperbolic geometry, space is negatively curved, which means it curves away from itself, creating a non-Euclidean geometry. This curvature is in direct contrast to Euclidean space, which is flat (zero curvature). The Poincaré ball is one of several models used to represent hyperbolic space. In this model, the entire hyperbolic space is mapped within the interior of a unit ball (i.e., a ball with a radius of 1). The boundary of the ball represents infinity in hyperbolic space.

*Geodesics* in the Poincaré ball model are the shortest paths or curves between two points within the unit ball, and the geodesic that passes through the center of the Poincaré ball between two points is the orthogonal bisector of the chord connecting those points. Unlike Euclidean geometry, where straight lines are the shortest paths, geodesics in hyperbolic geometry are curved and can be thought of as the most efficient routes between points within the curved space. Geodesics in hyperbolic space curve inwards towards the center of the Poincaré ball. Given two points within the unit ball, there is always one unique geodesic connecting them.

Geodesics in the Poincaré ball model are essential in various fields, including mathematics, physics, and computer science. They are used to study hyperbolic geometry, model complex networks, and capture hierarchical relationships in data and machine learning.

### C.0.2  GYROVECTOR SPACES

Gyrovector spaces are mathematical structures used in hyperbolic geometry to extend vector-like operations to spaces with constant negative curvature, such as hyperbolic space (Ungar, 2008; 2001; 2022). They are a fundamental concept within the framework of non-Euclidean geometry and provide a way to perform vector-like operations in these curved spaces.

Hyperbolic geometry, also known as Lobachevskian geometry, is a non-Euclidean geometry in which the curvature of space is constant and negative. Unlike Euclidean geometry, where parallel lines never intersect, hyperbolic geometry allows multiple parallels through a single point and exhibits various properties distinct from Euclidean space. In Euclidean geometry, vectors represent quantities that have both magnitude and direction. *Gyrovector spaces* aim to provide a similar framework for working with quantities in hyperbolic space. However, due to the curvature of this space, the behavior of vectors and vector-like entities differs from what we observe in Euclidean geometry.

A gyrovector space is based on the concept of a gyrogroup. A gyrogroup is a set equipped with a binary operation, analogous to addition in vector spaces. The operation in gyrogroups is often referred to as gyroaddition. Gyroaddition combines two elements of the gyrogroup to produce a result that retains the properties of a gyrovector. A *gyrovector space* is a vector space defined within the context of hyperbolic geometry. It consists of gyrovectors as its elements and gyroaddition as the binary operation. Gyrovector spaces obey the rules of vector spaces, such as closure, associativity, and the presence of additive and multiplicative identities.

