# OpenReview forum: "Generalizing Poincaré Policy Representations in Multi-agent Reinforcement Learning"
_ICLR.cc/2024/Conference — Submitted to ICLR 2024_

### Official Review · Reviewer_iHns · 2023-10-29

**Soundness:** 3 good
**Presentation:** 3 good
**Contribution:** 2 fair
**Rating:** 5
**Confidence:** 2

**Summary:**

The work addresses the problem of policy representation for multi-agent systems and does this by employing the so-called Poincare Policy representation. It provides motivations for the loss functions used to train a network in order to learn such representation and shows empirical evidence of the effectiveness of the proposed method.

**Strengths:**

The problem of policy representation is an interesting open problem, and this is particularly true for the multi-agent setting in which the joint policy has a potentially complex landscape. The presentation is rather clear and the empirical evidence is verified.

**Weaknesses:**

What I am mostly concerned about is the novelty of the work compared to the related works such as [1] where similar ideas were presented. I believe that the section describing the experimental setting could be substituted by a more in-depth description of how the proposed method should be employed in an RL training routine (Appendix A.1), and how it compares to the related works (Appendix A.3).

[1] Poincaré Embeddings for Learning Hierarchical Representations, Nickel, Kiela (2017) https://arxiv.org/pdf/1705.08039.pdf

**Questions:**

My only and most concerning question is related to the weaknesses I highlighted, namely:
- what is the novelty of the work compared to the related works in terms of the policy representation technique?

---

### Official Review · Reviewer_MnTe · 2023-10-30

**Soundness:** 2 fair
**Presentation:** 3 good
**Contribution:** 2 fair
**Rating:** 5
**Confidence:** 4

**Summary:**

This paper focuses on learning policy representations for multi-agent reinforcement learning and provides a new geometric policy representation perspective from the non-Euclidean hyperbolic projection. They leverage the hierarchical structure inherent in the Markow decision process and project the trajectories of multiple agents into Poincare ball. Concretely, to enhance the hierarchical property, they design a contrastive objective to encourage consistent policies to be embedded closer in the hyperbolic space, while pushing inconsistent policies farther apart. Finally, the experimental results on two multi-agent benchmarks showcase the superiority of the proposed method across cooperative and competitive games.

**Strengths:**

1. The problem this paper considers is rather important.
2. The literature review in Appendix A.3 seems to be sufficient.
3. The proposed method that involves hyperbolic embeddings for multi-agent representation is novel.
4. This paper is generally clear and easy to follow.

**Weaknesses:**

1. The motivation of the proposed method is not clear enough. Figure 1 is too blurry and the reviewer cannot understand the points of this figure. More importantly, the authors should explain why hierarchical structures are inherent to multi-agent MDPs.
2. The proposed method is not sound. The main components of this paper are two loss functions: consistent loss and discriminative loss. These two losses cluster the embeddings of the same policy and separate the embeddings of different policies. If the embeddings are learned only through these two losses, only 'agent IDs' are encoded in the embeddings.  What if one agent chooses different policies or stochastic policies? What if different agents adopt the same policies? More seriously, it seems that the embeddings are the representations of policies, rather than states. If so, why is hyperbolic space needed? What are the hierarchical structures of agent representations?
3. The empirical results are not significant enough. There is only a slight improvement over baselines in Figure 5. The results in Figure 8 cannot strongly support the effectiveness of the proposed two objectives also due to the minor improvement.
4. This paper did not provide any discussions of limitations.

**Questions:**

Some questions are listed above, and here are other questions.

1. Why do the learning curves in Figure 4 first increase and then decrease?
1. How does the proposed method combine with PPO?
1. What are the effects of hyperparameters such as $\epsilon, \beta$.
1. Is the representation learning conducted in the trajectory level instead of state level?

---

### Official Review · Reviewer_VeDg · 2023-11-01

**Soundness:** 2 fair
**Presentation:** 2 fair
**Contribution:** 2 fair
**Rating:** 3
**Confidence:** 2

**Summary:**

The authors propose to analyse policy representations using a geometric lens. For this purpose, they utilise the tree structure of trajectories to develop a Poincare ball based policy representation. This process is done for all agents in a multi-agent system with the objectives being that representations of policies of the same agent derived from their trajectories should be closes when compared with representations of policies of other agents. They term the representations learnt using this contrastive method as Poincare Policy Representations.

**Strengths:**

1. The geometric lens of looking at policy representations, specifically using hyperbolic geometry is an interesting approach.
2. The hyperbolic geometry background needed has been succinctly explained in the manuscript.

**Weaknesses:**

The analysis has been motivated by MDPs and extended to games. I do not think this extension is straightforward and I have highlighted my concerns in the questions section. Also, it is assumed that each agent can get the policy representations of other agents. I do not think this information requirement is justifiable in MARL

**Questions:**

1. I did not understand the statement in the Introduction that "... policy representations in MARL are crucial to realize cooperation among agents, improve the performance and efficiency...". My difficulty in understanding is this - policy representations need to be shared between the agents in order to achieve some of these benefits. But if this is done, then the information structure of the game is changed and potentially the equilibria as well. One can then ask the question - if additional information were to be shared between the agents to facilitate cooperation, will policy representations be the best information? Can the authors please clarify this?
2. Does the tree architecture described for trajectories for MDPs extend in a straightforward way to multi-agent systems with different information structures. Is it assumed that each agent sees a different tree from its own perspective?
3. I believe in equation (1), it is assumed that the policy is dependent only on the state (as this is sufficient for optimality in MDPs). But this is not the case in games. In addition to policies being history-dependent, games also have other considerations such as definition of the specific refinement of Nash equilibrium or team optimality as the solution concept. Hence, I do not think the formulation given in (1) seamlessly extends to multi-agent scenarios. Can the authors please clarify this?
4. Throughout the paper trajectories are assumed to be sequences of state-action pairs. However, this is not the case in games, where each agent has a different observation and hence different trajectory. Also, what is the structural assumption made about the policies used by the agents? This is not obvious, as in games, Markovian policies need not be optimal for any agent.
5. I am not aware of hyperbolic neural networks and hence my understanding is limited. Is the entire trajectory sequence used to predict the next element in a hyperbolic neural network (like an RNN) or just the current trajectory element is used to predict the next one? Or is there no such sequence prediction task defined here and the hyperbolic network is trained purely using the losses defined in equations (11) and (12)?
6. In equation (7) what are the state, action and observation spaces? Are the observations and actions for each agent assumed to lie in a Euclidean space? What about discrete categorical action spaces? Can they also be incorporated in this framework?
7. If two agents in a game follow identical policies, will they represented by the same point in the Poincare ball? If so, how?
8. I did not understand the first line in Section 3.3. Can the authors please explain?
9. Does the discriminative loss function in equation (12) also minimize the distance between the representations of two different policies from two different agents? Then, how does the contrastive learning work? Should there be a negative sign in equation (13) corresponding to this loss?
10. Can the authors explain how the learnt policy representations are used in MARL by the different agents?

---

### Official Review · Reviewer_SaBP · 2023-11-03

**Soundness:** 3 good
**Presentation:** 3 good
**Contribution:** 3 good
**Rating:** 6
**Confidence:** 3

**Summary:**

The paper proposes a multi-agent reinforcement learning method with novel policy representations. By modeling the interaction trajectories between agents and environments as a tree-growing process in a Poincare ball, an encoding network is learned to generate policy embeddings for each agent from its trajectories. Conditioned on this representation, an RL policy is trained. The proposed method shows improved performance in experiments of both cooperative and competitive games.

**Strengths:**

The paper is well written and the contribution is novel. The experiments are comprehensive with multiple baselines and different game settings.

**Weaknesses:**

The proposed method seems to be limited to discrete action space and a certain form of hyperbolic neural network. It would be good to add some explanations on whether the method could be extended to more general settings.

**Questions:**

Can the proposed method be extended to continuous action space and a more flexible form of hyperbolic neural networks?

---

### Meta-Review · Area_Chair_fSqH · 2023-12-05

**Metareview:**

(a) The paper proposes to extend the framework of Poincare representations to the multi-agent RL setting. They introduce 2 losses in addition - consistency for each agent, discrimination between agents. They apply it to the overcooked domain and the pommerman domain.

(b) the strengths are the intuitive appeal of the approach. The paper is reasonably well written, and the idea of using hyperbolic representations is interesting.

(c) the experiments are not super convincing, there are some questions about the soundness of the approach trivially extending from MDPs to games. Moreover this extension was not described clearly. The two additional losses seem somewhat heuristic as well.

(d) a more clear justification of the method and it's soundness, potentially even a proof of reach equilibrium. Moreover, more convincing experiments and a clear justification for the new additions would help.

**Justification For Why Not Higher Score:**

The experiments are not super convincing, there are some questions about the soundness of the approach trivially extending from MDPs to games. Moreover this extension was not described clearly. The two additional losses seem somewhat heuristic as well. Overall, the paper has the potential to be interesting but falls a bit short on explanation, motivation, and experiments.

**Justification For Why Not Lower Score:**

N/A

---

### Decision · Program_Chairs · 2024-01-16

Reject